# Impact of Psychological Distress and Sleep Quality on Balance Confidence, Muscle Strength, and Functional Balance in Community-Dwelling Middle-Aged and Older People

**DOI:** 10.3390/jcm9093059

**Published:** 2020-09-22

**Authors:** Raquel Fábrega-Cuadros, Agustín Aibar-Almazán, Antonio Martínez-Amat, Fidel Hita-Contreras

**Affiliations:** Department of Health Sciences, Faculty of Health Sciences, University of Jaén, 23071 Jaén, Spain; rfabrega@ujaen.es (R.F.-C.); amamat@ujaen.es (A.M.-A.); fhita@ujaen.es (F.H.-C.)

**Keywords:** fall risk, balance, muscle strength, anxiety, depression, sleep quality

## Abstract

The objective was to evaluate the associations of psychological distress and sleep quality with balance confidence, muscle strength, and functional balance among community-dwelling middle-aged and older people. An analytical cross-sectional study was conducted (*n* = 304). Balance confidence (Activities-specific Balance Confidence scale, ABC), muscle strength (hand grip dynamometer), and functional balance (Timed Up-and-Go test) were assessed. Psychological distress and sleep quality were evaluated by the Hospital Anxiety and Depression Scale and the Pittsburgh Sleep Quality Index, respectively. Age, sex, physical activity level, nutritional status, and fatigue were included as possible confounders. Multivariate linear and logistic regressions were performed. Higher values of anxiety (OR = 1.10), fatigue (OR = 1.04), and older age (OR = 1.08) were associated with an increased risk of falling (ABC < 67%). Greater muscle strength was associated with male sex and improved nutritional status (adjusted R^2^ = 0.39). On the other hand, being older and using sleeping medication were linked to poorer functional balance (adjusted R^2^ = 0.115). In conclusion, greater anxiety levels and the use of sleep medication were linked to a high risk of falling and poorer functional balance, respectively. No associations were found between muscle strength and sleep quality, anxiety, or depression.

## 1. Introduction

Aging brings with it a series of changes that can affect the mobility and independence of people [1]. These changes affect the mood and attitude towards their environment, and this depends largely on the degree of acceptance of aging since it contributes to the feeling of happiness and satisfaction with life, whose lack can cause feelings of loneliness and sadness [2].

Certain disorders associated with this process, such as anxiety and/or depression, are psychological indicators of a decrease in quality of life [3]. Specifically, the prevalence of depression in the geriatric population worldwide is 7%, and its incidence increases with age [4]. Conversely, the prevalence of anxiety in people over 60 years old ranges between 0.7% and 18.6%, values far below those of younger adults [5].

Sleep quality is a key contributor to good health, and its importance among the older population cannot be overstated, given that sleep disorders and the difficulty to fall asleep become more common with age [6]. It has been shown that although the need to sleep remains the same throughout an individual’s life, the ability to get enough sleep does in fact decrease with age. This brings about several adverse health outcomes such as reduced physical function, depression, increased risk of falls, and mortality [7].

Falls represent a major health care problem among older people and are linked to increased morbidity, mortality, and health costs [8]. Around 30% of older people living in the community experience a fall each year [9]. Fall risk factors have been studied in detail and include demographic, environmental, and health-related factors [10]. Balance confidence is one of the most important psychological factors linked to falls and the deterioration of balance, and its decrease can lead to diminished independence and participation in activities of daily living, thus creating a vicious circle that affects the quality of life and creates more isolated and dependent people [11]. On the other hand, the impaired functional balance has been shown to be one of the most important predictors of falls [12].

Muscle strength also declines with age more sharply than muscle mass [13]. It has been reported that muscle loss in older women decreases 3.7% per decade, however, strength decreases 15% per decade until age 70 when the loss accelerates considerably [14]. Moreover, in 2018 the European Working Group on Sarcopenia (EWGSOP2) listed low strength as a primary indicator of probable sarcopenia [15]. A decrease in muscle strength contributes to an elevated prevalence of falls and the loss of functional capacity and is a major cause of disability, mortality, and other adverse health outcomes [16].

Not many studies have examined the impact of psychological distress and sleep quality on balance confidence and function, and muscle strength in older people, which, in many cases, have shown inconclusive results and have focused on sleep duration or insomnia. Based on all of the above, the objective of this study was to evaluate the associations of psychological distress and sleep quality with the risk of falling according to balance confidence, functional balance, and muscle strength among community-dwelling middle-aged and older individuals.

## 2. Experimental Section

### 2.1. Study Design and Participants

An analytical cross-sectional study was conducted, to which end 315 community-dwelling middle-aged and older people were initially contacted and 304 finally took part. Recruitment was performed by contacting several senior centers from the Eastern Andalusia region. Prior to the beginning of the study, all participants provided their written informed consent. The research was approved by the Research Ethics Committee of the University of Jaén, Spain (NOV.18/2.TES), and was conducted in accordance with the Declaration of Helsinki, good clinical practices, and all applicable laws and regulations.

Community-dwelling ambulatory adults aged 50 years and older, able to understand and complete the required questionnaires and willing to give written informed consent to participate in the study were included in the protocol. Exclusion criteria were: conditions that limit physical activity, chronic and/or severe medical disease or any neuropsychiatric disorder that could influence their responses to the questionnaires.

### 2.2. Study Parameters

#### 2.2.1. Balance Confidence

The Activities-specific Balance Confidence scale (ABC) was used to assess balance confidence in the performance of activities of daily living [17]. This is a 16-item questionnaire that quantifies the level of confidence in performing a specific task without losing balance or becoming unsteady [18]. Each item score ranges from 0–100%, and the total score is obtained by summing the ratings (0–1600) and then dividing by 16. A higher percentage indicates a greater level of balance confidence. A score of <67% has been identified as a reliable means of predicting a future fall [19]. This cut-off was used to identify which participants were at high risk of falling.

#### 2.2.2. Muscle Strength

Muscle strength was assessed with an analog dynamometer (TKK 5001, Grip-A, Takei, Tokyo, Japan). Participants were required to apply their maximum handgrip strength three times with the dominant hand, each separated by 30 s. The maximal measured effort was regarded as their handgrip strength [20].

#### 2.2.3. Functional Balance

The Timed Up-and-Go (TUG) test [21] is a simple and valid method for predicting changes in functional balance in older adults [22]. It is a sensitive and specific measure for identifying community-dwelling adults who are at risk of falls [23]. In the TUG test, individuals rise from a seated position on a chair, walk three meters, turn around, return, and sit down again. The time required to complete this task was recorded.

#### 2.2.4. Sleep Quality

Sleep quality was assessed using the Pittsburgh Sleep Quality Index (PSQI) [24,25]. It comprises 19 self-rated questions and 5 more (only used for clinical purposes) to be completed by bedmates or roommates. The 19 items (ranged from 0–3) generate a total score and seven different domains or subscales (subjective sleep quality, sleep latency, sleep duration, habitual sleep efficiency, sleep disturbance, use of sleeping medication, and daytime dysfunction). Higher scores indicate poorer subjective sleep quality.

#### 2.2.5. Psychological Distress

The Hospital Anxiety and Depression Scale (HADS) is a self-administered questionnaire widely used to assess psychological distress in the general population [26,27]. This questionnaire contains 14 items, 7 related to anxiety symptoms, and 7 to depressive symptoms. Each item ranges from 0–3, and the total scores for both anxiety and depression range from 0 to 21, with higher scores indicating more severe symptoms.

#### 2.2.6. Fatigue Severity

In order to assess fatigue severity during the last 7 days, the Fatigue Severity Scale was used [28]. This questionnaire consists of 9 items (rated from 1–7) and produces a total score where larger values imply greater fatigue.

#### 2.2.7. Nutritional Status

The Mini Nutritional Assessment survey (MNA) was used to evaluate nutritional status [29,30]. It has 18 questions that include anthropometric measures, health status, dietary patterns, and subjective assessments of nutritional and health status. Higher scores indicate better nutritional status.

#### 2.2.8. Physical Activity Level

Physical activity level was assessed with the International Physical Activity Questionnaire-Short Form (IPAQ-SF) [31]. It consists of seven items that measure physical activity within three intensity levels (walking, moderate, and vigorous) during an average week. Physical activity was evaluated by combining the activity score of both moderate and vigorous-intensity activity (min/day) for each work and recreational activity domain. Responses were converted to Metabolic Equivalent Task minutes per week (MET-min / week) according to the scoring protocol.

### 2.3. Sample Size Calculation

For sample size calculation, at least 20 subjects per variable are required in the linear regression model [32], while a minimum of 10 subjects per variable was needed in the logistic regression model [33]. Since 15 possible predicting variables were considered (7 domains plus the total score of the PSQI, anxiety, depression, as well as physical activity level, nutritional status, fatigue, sex, and age as possible confounders), over 300 subjects were required for the purposes of our analysis. The final number of participants was 304.

### 2.4. Statistical Analysis

Continuous variables were described using means and standard deviations, and for categorical variables frequencies and percentages were used. The Kolmogorov–Smirnov test was performed to evaluate the normal distribution of the data. To analyze the differences between participants with and without risk of falling (ABC), Student’s t test (continuous independent variables), and the Chi-squared test (sex) were used. In order to analyze the independent associations, a multivariate logistic regression was performed. Those variables with significant individual associations (*p* < 0.05) were selected for the stepwise logistic regression model. The odds ratio (OR) can be considered as significant when the 95% confidence interval (CI) does not include 1.00. The Chi-squared and Hosmer–Lemeshow tests were conducted to assess the overall goodness-of-fit for each of the steps of the model, as well as for the final model. To explore the possible individual associations of muscle strength and functional balance with PSQI, HADS, FSS, MNA, and IPAQ-SF scores, as well as with age (independent variables), Pearson’s correlation was used. As for the analysis of the independent associations, the same procedure was applied, but using a stepwise multivariate linear regression. Functional balance and muscle strength were individually introduced as dependent variables in separate models. We first looked into the bivariate correlation coefficients, and any independent variables with significant associations (*p* < 0.05) were included in the multivariate linear regression. Adjusted R^2^ was used to calculate the effect size coefficient of multiple determination in the linear models. R^2^ can be considered insignificant when <0.02, small if between 0.02 and 0.15, medium if between 0.15 and 0.35, and large if >0.35 [34]. A 95% confidence level was used (*p* < 0.05). Data management and analysis were performed using the SPSS statistical package for the social sciences for Windows (SPSS Inc., Chicago, IL, USA).

## 3. Results

A total of 304 participants (72.04 ± 7.88 years) took part in this study. When studying the ABC score (23.42 ± 7.25), 24.01% of participants were at risk of falling. The analysis revealed (Table 1) that participants with an ABC score < 67 were individually associated with the largest values of anxiety (*p* = 0.002), depression (*p* = 0.001), fatigue (*p* = < 0.001), increased age (*p* < 0.001), and worse nutritional status (*p* = 0.002).

The multivariate logistic regression that looked into the risk of falls as assessed with the ABC score revealed several significant results. Higher values of anxiety (OR = 1.10, 95% CI = 1.02–1.18), fatigue (OR = 1.04, 95% CI = 1.02–1.06), and older age (OR = 1.08, 95% CI = 1.04–1.12) were independently associated with ABC scores < 67%. The Hosmer–Lemeshow test showed a good fit of the model (Chi-squared = 2.403, *p* = 0.966), which was able to classify correctly 78.29% of all participants at high risk of suffering a future fall, according to the ABC score (Table 2).

As for functional balance (9.86 ± 2.91 s) and muscle strength (19.43 ± 6.42 kg), the individual associations are shown in Table 3. Muscle strength was only associated with anxiety (*p* = 0.001), fatigue (*p* = 0.020), and nutritional status (*p* = 0.038), whereas poor functional balance was related to greater age (*p* < 0.001) and physical activity level (*p* = 0.035), as well as with the use-of-sleeping-medication domain in PSQI (*p* = 0.028). Regarding sex differences, men displayed greater muscle strength (both *p* < 0.001), but worse functional balance (*p* = 0.005).

Lastly, the linear regression analysis (Table 4) revealed that greater muscle strength was independently associated with the male sex (*p* < 0.001) and a better nutritional status (*p* = 0.001), and the effect size was large (adjusted R^2^ = 0.392). On the other hand, being older (*p* < 0.001) and the use of sleeping medication (*p* = 0.033) were linked to poorer functional balance, although the effect size was small (adjusted R^2^ = 0.115).

## 4. Discussion

The objective of this study was to evaluate the associations of psychological distress and sleep quality with balance confidence, functional balance, and muscle strength among community-dwelling middle-aged and older individuals. In our study, anxiety, fatigue, older age, and the use of sleeping medication were shown to be associated with the risk of falling and poorer functional balance. Muscle strength was associated with being male and nutritional status.

In general, balance confidence scores are able to predict perceived physical function and even mobility in older adults [35]. Similar to our own study, a previously published paper also employed regression models to find a significant association of anxiety with confidence in balance, while depression was shown to be associated with avoidance of activity [36]. A systematic review with meta-analysis found an association between balance confidence and anxiety [37], and a similar link was established between depression and level of physical activity [38]. Regarding the association of age with balance confidence, Medley et al. [39] reported that participants with low balance confidence were older than those who reported high balance confidence. In our study, only anxiety, age, and fatigue were independently associated with the balance confidence. To our knowledge, this is the first study to observe an association between confidence in balance and fatigue in healthy middle-aged and older people, although there are studies that demonstrate this association, but in people with some pathology [40,41].

Muscle strength plays an important role in the execution of many activities of daily living and is considered an indicator of functional decline among community-dwelling older adults [42]. Low grip strength is predictive of poor outcomes and indicative of prolonged hospital stays, increased functional limitations, poor quality of life, and death [43]. For example, it has been shown that people who have a high level of grip strength have a significantly lower fear of falling than those who show lower levels [44,45]. In addition, it has been observed that the strength of the abductor muscles can identify older adults at risk of falling [46]. A recent study looking into the association between falls and lower-limb strength failed to find any link at a one-year follow-up [47]. Our study found no associations whatsoever between muscle strength and sleep quality, and increased muscle strength was independently associated only with being male (as in previous studies by Buchman et al. [48]) and with improved nutritional status. Other authors have agreed before that a poor diet is significantly associated with lower muscle strength, but they also linked it to lower physical function, longer TUG test time, depression, and risk of falling [49], although the results of the present study should be interpreted with caution since they are correlations and a cause-effect relationship cannot be established. Some recent studies even recommend the intake of supplementary proteins given their significant effect in increasing muscle mass and strength among elderly people with sarcopenia [50]. We must consider, however, that disparities in the literature may be due to a variety of population ages, measurement methods, and educational and cultural levels, which may have a confounding effect.

Balance confidence contributes to functional mobility performance [39], and there seems to be a strong link between balance self-efficacy and function capabilities [51]. A study by Brandão et al. [52] identified an association between excessive daytime sleepiness and quality of life, and also characterized the profile of older adults with poor sleep quality. Sleep duration is associated with inflammation markers (serum interleukin-6, tumor necrosis factor α, and C-reactive protein) in older adults, and in turn with mortality [53]. Loss of functional balance, as measured by the TUG test, is known to be one of the first signs of aging and is considered a marker for general health that is strongly associated with the risk of mortality [54]. In our results, and as far as individual associations are concerned, higher age, poorer sleep quality (use of sleep medication), and decreased levels of physical activity were linked to lower TUG scores. However, in the multivariate analysis model, such associations only held for the first two variables (age and poor sleep quality). The results of a study conducted among women indicate that a shorter sleep duration increased wakefulness after sleep onset, and decreased sleep efficiency are risk factors for functional or physical impairment in older women [55].

There are some limitations to our study that must be acknowledged. Firstly, its cross-sectional design did not allow for the evaluation of causal relationships. Secondly, sleep quality was assessed using self-report methods, and therefore the influence of recall bias must be considered. Thirdly, our study was conducted among people from a specific geographic area, and any generalization of its results should be limited to individuals with characteristics similar to those of our population sample. Future studies should consider exploring prospective designs, employing objective sleep quality assessment methods (polysomnography or actigraphy), and applying them to a general population of older adults.

## 5. Conclusions

Among middle-aged and older Spanish people, greater levels of anxiety and fatigue, as well as older age were associated with an increased risk of falling (assessed with the Activities-specific Balance Confidence scale). No associations were found with sleep quality and depression. Greater muscle strength was associated with being male and having a better nutritional status. Finally, increased age and the use of sleeping medication were linked to poorer functional balance.

## Figures and Tables

**Table 1 jcm-09-03059-t001:** Individual differences according to the risk of falling.

		All Participants (*n* = 304)		Risk of Falling (ABC)				
				No (*n* = 231)		Yes (*n* = 73)		*p*-Value
		Mean	SD	Mean	SD	Mean	SD	
PSQI	Sleep quality	1.01	0.82	1.00	0.80	1.04	0.87	0.738
	Sleep latency	1.18	1.09	1.16	1.08	1.26	1.11	0.475
	Sleep duration	1.00	0.97	1.00	0.98	1.01	0.94	0.890
	Sleep efficiency	0.95	1.10	0.97	1.11	0.90	1.08	0.658
	Sleep disturbances	1.23	0.58	1.22	0.57	1.27	0.61	0.461
	Use of sleeping medication	1.01	1.34	1.02	1.34	0.97	1.34	0.804
	Daytime dysfunction	0.49	0.63	0.46	0.61	0.59	0.68	0.130
	Total score	6.87	4.70	6.82	4.58	7.04	5.11	0.725
Anxiety		5.74	4.02	5.34	4.07	7.01	3.58	0.002
Depression		4.99	3.44	4.62	3.38	6.14	3.39	0.001
Physical activity level, MET-min/week		1367.96	2213.43	1310.16	1817.89	1549.29	3157.97	0.422
Nutritional status		26.31	2.08	26.52	1.99	25.64	2.22	0.002
Fatigue		21.33	15.25	18.87	13.96	29.14	16.58	<0.001
Age, years		72.04	7.88	70.99	7.27	75.38	8.82	<0.001
		*n*	%	*n*	%	*n*	%	
Sex	Male	255	83.88	37	16.02	12	16.44	0.932
	Female	49	16.12	194	83.98	61	83.56	

ABC: Activities-Specific Balance Confidence Scale. MET: Metabolic Equivalent of Task. PSQI: Pittsburgh Sleep Quality Index. SD: Standard Deviation.

**Table 2 jcm-09-03059-t002:** Multivariate logistic regression analyses for factors associated with the risk of falling (determined through the ABC score).

		OR	95% CI	*p*-Value
Risk of falling (ABC)	Anxiety	1.10	1.02–1.18	0.012
	Fatigue	1.04	1.02–1.06	0.000
	Age	1.08	1.04–1.12	0.000

ABC: Activities-Specific Balance Confidence Scale. CI: Confidence Interval. OR: Odds Ratio.

**Table 3 jcm-09-03059-t003:** Pearson’s correlations of functional balance and muscle strength, with PSQI scores and possible confounders.

		Muscle Strength		Functional Balance	
		r	*p*-Value	r	*p*-Value
PSQI	Sleep quality	0.06	0.264	0.03	0.654
	Sleep latency	0.07	0.231	0.00	0.950
	Sleep duration	0.05	0.427	0.02	0.790
	Sleep efficiency	−0.00	0.963	−0.02	0.747
	Sleep disturbances	0.01	0.808	−0.02	0.665
	Use of sleeping medication	0.03	0.602	0.13	0.028
	Daytime dysfunction	0.01	0.854	0.04	0.474
	Total score	0.05	0.416	0.04	0.464
Anxiety		−0.18	0.001	0.05	0.355
Depression		−0.08	0.191	0.08	0.174
Nutritional status		−0.12	0.038	−0.05	0.401
Fatigue		−0.14	0.017	0.06	0.281
Age, years		−0.104	0.070	0.33	0.000
Physical activity level (MET-min / week)		0.103	0.075	−0.12	0.035

MET: Metabolic Equivalent of Task. PSQI: Pittsburgh Sleep Quality Index. r: Pearson’s Correlation Coefficient.

**Table 4 jcm-09-03059-t004:** Multivariate linear regression analyses for functional balance and muscle strength.

		B	β	95% CI		*p*-Value
Muscle strength	Sex	−10.74	−0.62	−12.28	−9.21	<0.001
	Nutritional status	0.44	0.14	00.17	0.71	0.002
Functional balance	Age	0.12	0.34	−0.09	−0.16	<0.001
	Use of sleeping medication	0.27	0.12	0.04	0.50	0.023

B: Unstandardized Coefficient. β: Standardized Coefficient. CI: Confidence Interval. MET: Metabolic Equivalent of Task. PSQI: Pittsburgh Sleep Quality Index.

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
