# Peer review of "Impact of Psychological Distress and Sleep Quality on Balance Confidence, Muscle Strength, and Functional Balance in Community-Dwelling Middle-Aged and Older People"

_jcm, 2020, doi:10.3390/jcm9093059_

Round 1
Reviewer 1 Report
Generally well clear with some interesting findings to support a relationship between anxiety, fatigue and falls.
Minor comments:
Page 1 introduction: "The experience of aging brings about wide ranging effects on the body, with damaging effects on the mobility and independence of individuals [1]. This in turn renders people more vulnerable to alterations in their mental state due to the need to adapt to new conditions." It is a bit unclear exactly what this means.
The description of the various factors associated with falls and quality of life is on the whole clear. However, the rationale for linking these issues if not fully described in the introduction and reflected in the aims / objectives. The key question being - what makes this research unique?
It would be helpful if the objectives were a little clearer about what the study would do (i.e. to analyse the relationship between balance and XX variables but also to explore predictors of falls risk (defined by a score of <67 in the ABC). In fact the results were presented in the reverse order to the data analysis written methods which adds to this ambiguity.
Discussion:
"Some recent studies even recommend the intake of supplementary proteins given their significant effect in increasing muscle mass and strength among elderly people with sarcopenia". It may be worth adding the caution with interpreting cause and effect from correlational findings.
The discussion mostly discussed the correlational findings and not the factors associated with falls - which were the first reported finding in the results and the focus of the conclusion.
General comments:
The paper would benefit from a more focused approach to the objectives and presentation and discussion of the findings (see comments above for more details). Otherwise, methods and results tables presented clearly.
Author Response
Generally well clear with some interesting findings to support a relationship between anxiety, fatigue and falls.
We very much appreciate your constructive comments, useful information and your time. Responses to your comments are written in bold and changes made in the manuscript are marked in red.
Minor comments:
Page 1 introduction: "The experience of aging brings about wide ranging effects on the body, with damaging effects on the mobility and independence of individuals [1]. This in turn renders people more vulnerable to alterations in their mental state due to the need to adapt to new conditions." It is a bit unclear exactly what this means.
We want to thank the reviewer for this comment. The first paragraph of the introduction section has been rearranged (lines 29-32).
The description of the various factors associated with falls and quality of life is on the whole clear. However, the rationale for linking these issues if not fully described in the introduction and reflected in the aims / objectives. The key question being - what makes this research unique?
Thank you very much for this suggestion. In order to follow it, new text has been included in the last paragraph of the introduction section (lines 59-61).
It would be helpful if the objectives were a little clearer about what the study would do (i.e. to analyse the relationship between balance and XX variables but also to explore predictors of falls risk (defined by a score of <67 in the ABC). In fact the results were presented in the reverse order to the data analysis written methods which adds to this ambiguity.
We thank the reviewer for this comment. The last paragraph of the introduction and the statistical analysis sections have been rearranged (lines 63 and 139-155).
Discussion:
"Some recent studies even recommend the intake of supplementary proteins given their significant effect in increasing muscle mass and strength among elderly people with sarcopenia".It may be worth adding the caution with interpreting cause and effect from correlational findings.
Thank you very much for this suggestion, the discussion section has been modified accordingly (lines 234-236)
The discussion mostly discussed the correlational findings and not the factors associated with falls - which were the first reported finding in the results and the focus of the conclusion.
In order to follow the reviewer’s suggestion, more information and references has been included in the second dparagraph of th s¡discussion section (lines 215-221).
General comments:
The paper would benefit from a more focused approach to the objectives and presentation and discussion of the findings (see comments above for more details). Otherwise, methods and results tables presented clearly.
Thank you very much for these comments. It is our belief that the manuscript is substantially improved after making the suggested edits.
Reviewer 2 Report
This manuscript describes a cross-sectional study among community-dwelling middle-aged and older persons. The results were modest, however, I have some comments that might improve the quality of the manuscript.
- In the introduction “muscle strength also decreases with age, causing even greater difficulties for older individuals than the decrease in muscle mass” This sentence is very vague and without any citations to support. Please add a citation and examples.
- Please use the term ”functional mobility” not “functional balance” as the main term for the Tug Test.
- Please clarify the statistical analysis.
- The structure in the text does not fit with the order of the tables (they are mixed), please correct this.
- “Pearson’s correlation was used to explore the possible associations of muscle strength, functional balance, and waist circumference with PSQI, HADS, FSS, MNA, and IPAQ-S scores, as well as with age (independent variables)”. Please explain why is the waist circumference being mentioned here suddenly? Must be a clerical error
- Table 1. Functional balance and handgrip strength were presented as a central component of this research, therefore, there was no information in table 1. So the reader faces incomplete information about the role of handgrip strength and functional mobility to analyze the differences between participants with and without risk of falling.
- In Table 3 why the variable ABC scale was not taken into account. Because of the multicollinearity? Please clarify!
- In the results section
- “The multivariate logistic regression that looked into the risk of falls as assessed with the ABC and FES-I scores revealed several significant results”. Please explain why the Fes-i is being mentioned here suddenly?
- “Muscle strength was only associated with anxiety and fatigue (p = 0.020),” . The nutritional status is still missing here (p- value 0.038) and please add the p-values for each variable.
- “Lastly, the linear regression analysis (Table 4) revealed that greater muscle strength was independently associated with male sex (p < 0.001) and a better nutritional status (p = 0.001), and the effect size was large (adjusted R2 = 0.392)”. Do you mean here the beta coefficient was large?
- In the discussion section
- “In our study, only anxiety, age, and fatigue were significantly associated with balance confidence in our community-dwelling middle-aged and older population”. According to Table 1 also depression and nutritional status associated with balance confidence. Please clarify!
- The authors explain at the beginning an association with handgrip strength and various variables, and then they bring examples with lower limb strength. It would be ideal to give examples with handgrip strength and fear falling.
Author Response
This manuscript describes a cross-sectional study among community-dwelling middle-aged and older persons. The results were modest, however, I have some comments that might improve the quality of the manuscript.
We very much appreciate your constructive comments, useful information and your time. Responses to your comments are written in bold and changes made in the manuscript are marked in red.
- In the introduction “muscle strength also decreases with age, causing even greater difficulties for older individuals than the decrease in muscle mass” This sentence is very vague and without any citations to support. Please add a citation and examples.
We want to thank the reviewer for this comment. In order to follow it the manuscript has been modified accordingly (lines 52-54).
- Please use the term ”functional mobility” not “functional balance” as the main term for the Tug Test.
We thank the reviewer for this comment. We agree with the reviewer that it is well established that the TUG test is used to assess functional mobility, but, as mentioned in the methods (Experimental Section), it is also appropriate to evaluate functional balance in older adults. The experimental section (2.2. Study parameters, lines 96-97) has been modified and one reference has been added:
Benavent-Caballer V, Sendín-Magdalena A, Lisón JF, Rosado-Calatayud P, Amer-Cuenca JJ, Salvador-Coloma P, Segura-Ortí E. Physical factors underlying the Timed "Up and Go" test in older adults. Geriatr Nurs. 2016;37(2):122-7. doi: 10.1016/j.gerinurse.2015.11.002.
- Please clarify the statistical analysis.
We thank the reviewer for this comment. The statistical analysis section has been rearranged (lines 139-155).
- The structure in the text does not fit with the order of the tables (they are mixed), please correct this.
Thank you very much. The structure of the text is in accordance with tables. Tables 1 and 2 present the individual and independents associations, respectively with the risk of falling according to the ABC score, and the associations with muscle strength and functional balance are shown in table 3 (individual) and table 4 (independent).
- “Pearson’s correlation was used to explore the possible associations of muscle strength, functional balance, and waist circumference with PSQI, HADS, FSS, MNA, and IPAQ-S scores, as well as with age (independent variables)”. Please explain why is the waist circumference being mentioned here suddenly? Must be a clerical error
We want to apologize, it was effectively a clerical error and has been removed.
- Table 1. Functional balance and handgrip strength were presented as a central component of this research, therefore, there was no information in table 1. So the reader faces incomplete information about the role of handgrip strength and functional mobility to analyze the differences between participants with and without risk of falling.
We want to thank the reviewer for this comment. In the results section we have followed the order presented in the last paragraph of the introduction section, and the first outcome was the balance confidence (concretely the risk of falling according balance confidence), followed by muscle strength and functional balance. Information regarding handgrip strength and TUG test (mean ± SD) has been included (line 181).
- In Table 3 why the variable ABC scale was not taken into account. Because of the multicollinearity? Please clarify!
We thank this comment. The ABC score was employed to classify participants as yes/no “risk of falling” according to balance confidence, which has been showed din tables 1 and 2. The mean ± SD of the ABC total score has been included in the first paragraph of the results section (line 160).
- In the results section
- “The multivariate logistic regression that looked into the risk of falls as assessed with the ABC and FES-I scores revealed several significant results”. Please explain why the Fes-i is being mentioned here suddenly?
We are sorry, this was an error and it has been removed.
- “Muscle strength was only associated with anxiety and fatigue (p = 0.020),” . The nutritional status is still missing here (p- value 0.038) and please add the p-values for each variable.
Thank you very much for this suggestion. The manuscript has been modified accordingly (lines 182-183).
- “Lastly, the linear regression analysis (Table 4) revealed that greater muscle strength was independently associated with male sex (p < 0.001) and a better nutritional status (p = 0.001), and the effect size was large (adjusted R2 = 0.392)”. Do you mean here the beta coefficient was large?
Adjusted-R2 was used to assess size effect, which was considered large when > 0.35 (as mentioned in the Statistical Analysis section).
- In the discussion section
- “In our study, only anxiety, age, and fatigue were significantly associated with balance confidence in our community-dwelling middle-aged and older population”. According to Table 1 also depression and nutritional status associated with balance confidence. Please clarify!
We want to thank the reviewer for this comment. Effectively, depression and nutritional status were associated with the risk of falls according to the balance confidence (table 1), but these associations disappeared in the linear regression (independent associations). In order to avoid confusion, we have “significantly” was replaced by “independently” (line 218).
- The authors explain at the beginning an association with handgrip strength and various variables, and then they bring examples with lower limb strength. It would be ideal to give examples with handgrip strength and fear falling.
Thank you very much for this suggestion. The manuscript has been modified accordingly (page 7, lines 225-227).
Round 2
Reviewer 2 Report
After reading the revised version, I believe that the authors have addressed most of my comments and the manuscript has been significantly improved.